# How to assemble a scale-invariant gradient

**Arnab Datta, Sagnik Ghosh\*, Jane Kondev**

Department of Physics, Brandeis University, Waltham, United States

**Abstract** Intracellular protein gradients serve a variety of functions, such as the establishment of cell polarity or to provide positional information for gene expression in developing embryos. Given that cell size in a population can vary considerably, for the protein gradients to work properly they often have to be scaled to the size of the cell. Here, we examine a model of protein gradient formation within a cell that relies on cytoplasmic diffusion and cortical transport of proteins toward a cell pole. We show that the shape of the protein gradient is determined solely by the cell geometry. Furthermore, we show that the length scale over which the protein concentration in the gradient varies is determined by the linear dimensions of the cell, independent of the diffusion constant or the transport speed. This gradient provides scale-invariant positional information within a cell, which can be used for assembly of intracellular structures whose size is scaled to the linear dimensions of the cell, such as the cytokinetic ring and actin cables in budding yeast cells.

## Editor's evaluation

How biological patterns such as concentration gradient scale with the size of the cell or organism is a long-standing question in developmental and cell biology. In this study, Datta et al., show theoretically that directed membrane transport of biomolecules and their release at the cell pole results in a cytoplasmic gradient that scales with cell size if two requirements are met: the cell grows while maintaining its spheroid proportions, (i.e. not by elongation), and the binding of the cytoplasmic fraction of the biomolecule to the membrane should be close to irreversible. While there are no experiments available to date to directly probe the proposed mechanism, it could be achieved through several biochemical implementations and can inspire experimental studies.

## Introduction

The living cell is not a well-mixed bag of chemicals. Different parts of the cell have different chemical composition and these spatial inhomogeneities are critical to life. The textbook example of this comes from the study of early fly development where spatial patterns of gene expression provide the information used by the embryo to set up the animal's body plan (*Gregor et al., 2005*). While the early embryo is a syncytium hundreds of microns in size, spatial patterns of proteins can also be observed in much smaller cells, like for example in budding yeast, where they drive the establishment of polarity (*Chiou et al., 2017*).

Self-assembly of structures within the cell can also exhibit spatial patterns. For example, cell division is typically facilitated by the formation of a cytokinetic ring, a multicomponent protein structure that assembles at a specific location in the dividing cell. In cells that divide symmetrically, the cytokinetic ring self-assembles in the middle of the cell. This observation raises the intriguing question, how does the cell 'know' where its middle is? In the case of *Escherichia coli*, pole-to-pole oscillations of the Min proteins provide information about its middle and guide the assembly of the cytokinetic ring (*Ramm et al., 2019*). Budding yeast cells on the other hand divide asymmetrically, by assembling the cytokinetic ring close to the cell pole. Remarkably, recent experiments have found that yeast cells of

**\*For correspondence:**
sagnik@brandeis.edu

**Competing interest:** The authors declare that no competing interests exist.

different size assemble cytokinetic rings of different size, such that the diameter of the ring scales with the diameter of the cell (**Kukhtevich et al., 2020**). This scaling property implies that the self-assembly of the cytokinetic ring occurs at a location that is at a fixed relative distance from the pole. In other words, the location of the ring is specified not in absolute units of distance but in units relative to the cell diameter.

The process of assembling a cytokinetic ring is an example of a common engineering problem faced by cells: how to set up a coordinate system that specifies positional information? This engineering challenge is usually met by the formation of an intracellular gradient, whereby the concentration of a molecular species varies across the dimensions of the cell. In the presence of such a gradient, chemical reactions, such as those leading to the formation of the cytokinetic ring, can be localized to specific region of the cell.

Intracellular gradients have been shown to be involved in a variety of processes in cells (**Folkmann and Seydoux, 2018**; **Reber and Goehring, 2015**; **Hubatsch and Goehring, 2020**). A well-studied example is provided by the aforementioned Bicoid gradient in the fruit fly embryo, which plays a role in patterning gene expression along the length of the cell (**Gregor et al., 2005**; **Grimm et al., 2010**). The Bicoid protein is synthesized at the anterior of the embryo from maternally deposited Bicoid mRNA localized to the anterior pole. The newly synthesized protein diffuses throughout the cell and it degrades over time, leading to a protein gradient with a higher concentration near the source. Similarly, the MEX-5 gradient in the *Caenorhabditis elegans* zygote (**Wu et al., 2018**) is involved in asymmetric division of the cell. MEX-5 exists in two phosphorylation states, one which is fast diffusing (FD) and the other is slow diffusing (SD). While the FD species is present uniformly throughout the cell, the switch from FD to SD is faster at the anterior end resulting in a gradient of SD MEX-5. Another well-studied protein gradient is Pom1 in fission yeast (**Allard et al., 2019**; **Moseley et al., 2009**) which controls cell division at the middle of the cell. Like MEX-5, Pom1 exists in two states, an SD one that is bound to the membrane and an FD state in the cytoplasm. Pom1 is recruited to the cell surface near the cell poles, and this results in a higher membrane-bound concentration near the pole than the middle of the cell.

In the fruit fly embryo, the Bicoid protein, which is produced from Bicoid mRNA localized at the anterior pole, freely diffuses in the cytoplasm and undergoes degradation with a lifetime $\tau$. The combined effect of protein production, diffusion, and degradation leads to a concentration of Bicoid that varies along the anterior-posterior axis. The dynamics of the Bicoid gradient can be described by a one-dimensional reaction-diffusion equation (**Gregor et al., 2005**):

$$\frac{\partial c}{\partial t} = D\frac{\partial^2 c}{\partial^2 z} - \frac{c}{\tau}. \tag{1}$$

The steady-state solution to this equation is an exponentially decaying concentration gradient, $c\left(z\right) = c_0\,e^{-z/\lambda}$ where $\lambda = \sqrt{\tau D}$ is the decay length. Here, $c\left(z\right)$ is the concentration of Bicoid along the anterior-posterior ($z$) axis and $D$ is the diffusion constant of Bicoid. Embryos of different fly species have Bicoid gradients that are scaled to their linear dimensions, which can vary by an order of magnitude (**Gregor et al., 2005**). This observation provides a key motivation for the theoretical investigation herein, of a mechanism that can produce such a scale-invariant intracellular gradient.

Different mechanisms for assembling scale-invariant gradients have been previously discussed (**Ben-Zvi et al., 2011b**). They typically start with **Equation 1** and consider ways in which scaling can be achieved by effectively making the diffusion rate, or the lifetime of the protein, sensitive to the size of the cell in which the gradient is established. For example, one idea is that the protein degradation in the fly embryo occurs in the nuclei and therefore the effective degradation rate is proportional to the density of nuclei (**Gregor et al., 2007**). Regardless of embryo size, the number of nuclei, which in the early fly embryo are located at its surface, is fixed at a given stage of development, and therefore their number density scales inversely with the surface area $\sim 1/L^2$ ; here, $L$ is the linear dimension of the embryo. This relation leads to an effective protein lifetime $\tau \sim L^2$ and therefore to linear scaling of the decay length with embryo length, $\lambda = \sqrt{\tau D} \sim L$.

Other mechanisms that employ two diffusing morphogens that are produced from opposing ends of the cell (or tissue) have been proposed. These mechanisms produce scaling of the gradient from the interactions of the two morphogens, which activate and suppress each other (**Ben-Zvi et al., 2011b**). For example, in the expander-repressor model (**Ben-Zvi and Barkai, 2010**), the interactions between the repressor, produced at the anterior pole, and the expander, produced at the posterior

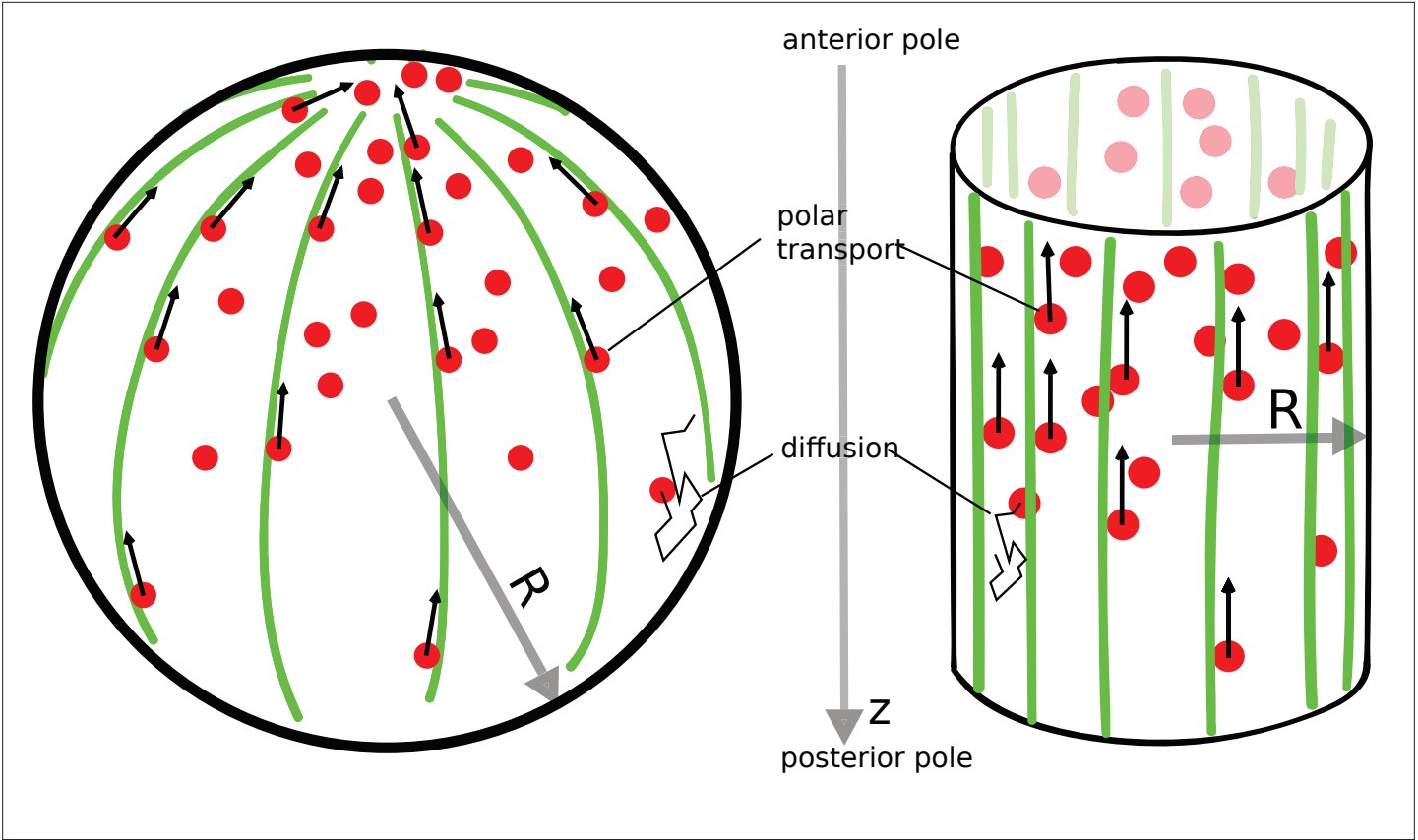

**Figure 1.** The polar transport model of gradient formation. Transport of proteins (red) in the cell is a combination of diffusion and polar transport. Diffusion occurs throughout the cytoplasm. When diffusing proteins encounter motor proteins (black arrows) moving along polar filaments (green) that contour the cell surface, they can be taken up by the motors and delivered to the cell's anterior pole. The anterior pole acts as a source, while the filaments along the surface of the cell serve as a sink of diffusing proteins. The result of this combined polar transport and diffusion is a protein gradient that extends along the polar (z) axis of the cell, with proteins accumulating toward the anterior pole.

pole, fix the repressor concentration $c\left(\frac{z}{\lambda}\right)$ at the posterior pole, that is, $c\left(\frac{L}{\lambda}\right) = const$. This then leads to scaling of the decay length of the repressor gradient $\lambda$ with the distance between the two poles $L$. It has been argued that this mechanism is responsible for the Dpp activation gradient in *Drosophila* wing imaginal discs (***Ben-Zvi et al., 2011a***).

Here, we propose a mechanism of gradient formation that makes use of directed transport of proteins along the surface of the cell. This type of cortical transport is known to occur in early embryos of *C. elegans* due to the contractility of the cortical layer of actin and myosin (***Mayer et al., 2010***), but also in much smaller cells, like budding yeast with its polar transport along actin cables confined to the cell surface (***Chesarone-Cataldo et al., 2011***), or the motor-driven transport along microtubules in cilia, such as in *Chlamydomonas* cells (***Ishikawa and Marshall, 2011***). Our key result is that directed transport of proteins along the cell cortex to the cell pole, when coupled to diffusion in the cell interior, produces a scale-invariant gradient. We show that the shape of the gradient is solely determined by the linear dimensions of the cell. Namely, for biologically realistic parameters that correspond to efficient capture of proteins by the surface transport, the gradient is independent of the diffusivity of the proteins and their transport speed, as well as the chemical rate constants that describe the binding and unbinding of the proteins from the molecules involved in their transport.

## Results

### Polar transport model of gradient formation

In a variety of cells (fission yeast [**Kovar et al., 2011**], budding yeast [**Chesarone-Cataldo et al., 2011**], *Drosophila* oocyte [**Raman et al., 2018**]), there exists a polar arrangement of cytoskeleton filaments, leading to transport of proteins toward the anterior pole, as illustrated in **Figure 1**. Following the example of actin cables in budding yeast cells (**Chesarone-Cataldo et al., 2011**), we consider a network of filaments that is localized to the surface of the cell. Proteins that diffuse in the cell's cytoplasm can be captured by molecular motors moving along the filaments toward the anterior pole, where they are released (protein release before it reaches the pole is considered in a later section). This combined process of polar transport and cytoplasmic diffusion leads to a protein gradient, with an accumulation of the proteins at the anterior cell pole. The key result of this paper is that the gradient formed by this 'polar transport' mechanism is scale-invariant and controlled solely by the geometry of the cell.

To understand the geometric nature of the gradient, we can employ a simple argument that relates the gradient formed by the polar transport mechanism to the diffusion-degradation model (**Equation 1**). Instead of protein degradation proteins are removed from the cytoplasm when they are captured by the motors moving along the cell's cortex. The typical time for a protein diffusing in the cytoplasm to reach the cell's cortex is $\tau_D \approx R^2/D$, where $R$ is the radius of the cell (**Figure 1**) and $D$ is the diffusion constant of the proteins in the cytoplasm. If we think of $\tau_D$ as the lifetime of a protein in the cell, before it is captured by the motor proteins, we can use the diffusion-degradation model (**Equation 1**) to describe the concentration of proteins in the cytoplasm. In this case, the steady-state gradient is exponential, as in the diffusion-degradation model, with a length scale of spatial decay from the anterior pole given by $\lambda \approx \sqrt{D\tau_D} = R$. In other words, the length scale that defines the gradient is given by the radius of the cell and is independent of the diffusion constant, or the speed of motor transport. The gradient scales with the linear dimensions of the cell thereby providing positional information that is independent of cell size, similar to what is observed for the Bicoid gradient in embryos of different fly species (**Gregor et al., 2005**).

This simple calculation based on **Equation 1** ignores details of the cell shape and it assumes that every protein reaching the surface of the cell is captured and transported to the pole. Next, we address these shortcomings and show that a more careful analysis does not fundamentally change the main conclusion, that combined cytoplasmic diffusion and cortical transport of proteins to the cell pole leads to a scale-invariant protein gradient.

To compute the protein gradient produced by different cell shapes, we consider only shapes with azimuthal symmetry, namely, spherical, spheroidal, and cylindrical. The dynamics of the concentration of the protein in the cytoplasm ($c$) is described by a set of partial differential equations that describe protein diffusion, capture, and transport by surface-bound motors, and release at the anterior pole of the cell.

The first equation,

$$\frac{\partial c}{\partial t} = D\nabla^2 c + j(t)\,\delta(\mathbf{r} - \mathbf{r}_0) \tag{2.1}$$

describes the diffusion of proteins in the cytoplasm, with a source term that accounts for the release of proteins, which are transported along the surface, at the anterior pole. Proteins are released slightly off the pole, at $\mathbf{r}_0 = \mathbf{r}_{anterior\ pole} - (0, 0, \epsilon)$, where $\epsilon \ll$ size of the cell. (Note, in the case of the sphere, the coordinates for the two cell poles are $\mathbf{r}_{anterior\ pole} = (0, 0, R)$ and $\mathbf{r}_{posterior\ pole} = (0, 0, -R)$.) This shift off pole is required to make the partial differential equations well defined, and it has a negligible effect on the protein gradient.

The source term in **Equation 2.1** is, by continuity, equal to the protein current (number of proteins per unit time)

$$j(t) = 2\pi v \lim_{z \to z_{anterior\ pole}} c_s(z, t)\,\rho(z) \tag{2.2}$$

that are delivered to the release point by action of motors moving with speed $v$. Here, $c_s(z, t)$ is the protein concentration on the surface of the cell at position $z$ along the anterior-posterior axis, while $\rho(z)$ is the radius of the cell at that same position. To obtain the protein current at the release point,

we consider the current through a circle of perimeter $2\pi\,\rho\,(z)$ and take the limit when this circle shrinks to a point that is the anterior pole.

The third and final equation,

$$\frac{\partial c_s}{\partial t} = -D\,\nabla c\big|_{at\ surface} \cdot \hat{n} - \boldsymbol{v} \cdot \nabla c_s, \tag{2.3}$$

where $n$ is the unit vector normal to the cell surface, describes the directed transport of the proteins on the surface, by motors moving with velocity $\boldsymbol{v}$ along the polar direction of the cell. The first term in this equation is a source term that describes the diffusive encounter of the surface-bound motors with proteins in the cytoplasm. A key assumption here, which we reconsider later, is that every evert diffusional encounter of the protein with the surface leads to capture by a motor.

These partial differential equations come with two boundary conditions: (i) To account for the assumption that all proteins diffusing in the cytoplasm, when they reach the cell cortex, are taken up by the motors, we set the cytoplasmic concentration $c$ to zero at the surface of the cell, that is, $c\left(\boldsymbol{r_{surface}}, t\right) = 0$. (ii) The surface concentration $c_s$ is set to zero at the posterior pole, $c_s\left(\boldsymbol{r_{posterior\ pole}}, t\right) = 0$ to account for the fact that there is no influx of proteins from any other point on the surface to the posterior pole.

Since proteins are only captured and transported by motors moving along the filaments lining the cell surface, that is no proteins are produced or degraded over the time when the gradient is observed, the total number of proteins is assumed fixed, $\int_{volume} c + \int_{surface} c_s = N$.

In steady state, $j\left(t\right) = j$ is time independent, and since we are only interested in the concentration gradient of proteins diffusing in the cytoplasm, *Equation 2.1* simplifies to

$$\nabla^2 c = -q\delta\left(\boldsymbol{r} - \boldsymbol{r}_0\right), \tag{3}$$

where $q = \frac{j}{D}$ and the boundary condition $c\left(\boldsymbol{r_{surface}}\right) = 0$ accounts for the cytoplasmic proteins being taken up by the cortical transport. To compute the steady-state gradient, we note that *Equation 3* for the cytoplasmic concentration is the same as the Poisson equation in electrostatics with $q = \frac{j}{D}$ playing the role of a point charge, and the concentration $c$ is the analogue of the electrostatic potential. This equation automatically satisfies protein-number conservation as the source term balances the rate at which proteins are taken up at the surface due to Gauss' theorem.

Next, we consider different cell geometries. Using a combination of simulations and analytic calculations, we show that the protein gradient formed by the polar transport model is determined solely by the cell geometry and scales with the linear dimensions of the cell.

## Gradients in spherical cells of different size

We start by examining the simplest cell shape, a sphere. The direction of transport is toward the anterior pole. In *Figure 2A*, we show a snapshot of the steady-state distribution of proteins inside the cell that results from the combined polar transport and diffusion, obtained from direct simulations of the particle dynamics. As expected, proteins concentrate at the anterior pole of the cell. *Figure 2B* details how the shape of the gradient changes as we change the radius of the cell, while keeping the total concentration of proteins fixed. Changing the diffusion constant and velocity of polar transport only affects the fraction of particles that are on the surface of the cell versus the cytoplasm. In particular, it does not change the shape of the cytoplasmic gradient, which is what we focus on herein. The bump in the concentration profile at very short distances is due to the release point being placed at $\epsilon = 0.05R$ away from the anterior pole.

For the spherical cell, *Equation 3* is the analogue of the electrostatic problem of a conducting spherical shell at zero potential with a charge at $\boldsymbol{r}_0$, which can be solved by placing an image charge $-\frac{R}{r_0}q$ at $\boldsymbol{r_{out}} = \left(0, 0, \frac{R^2}{r_0}\right)$. Note that this is why the release point of proteins transported to the cell pole is placed slightly inside the cell, so that the continuum equations are well defined. In the electrostatic analogy, the Coulomb potential of the charge and its image combine to give the total potential, which solves the Poisson equation with the specified boundary condition. Therefore, the cytoplasmic concentration of the protein is given by

$$c\left(\boldsymbol{r}\right) = \frac{q}{4\pi\left|\boldsymbol{r} - \boldsymbol{r}_0\right|} - \frac{R}{r_0}\frac{q}{4\pi\left|\boldsymbol{r} - \boldsymbol{r_{out}}\right|}. \tag{4}$$

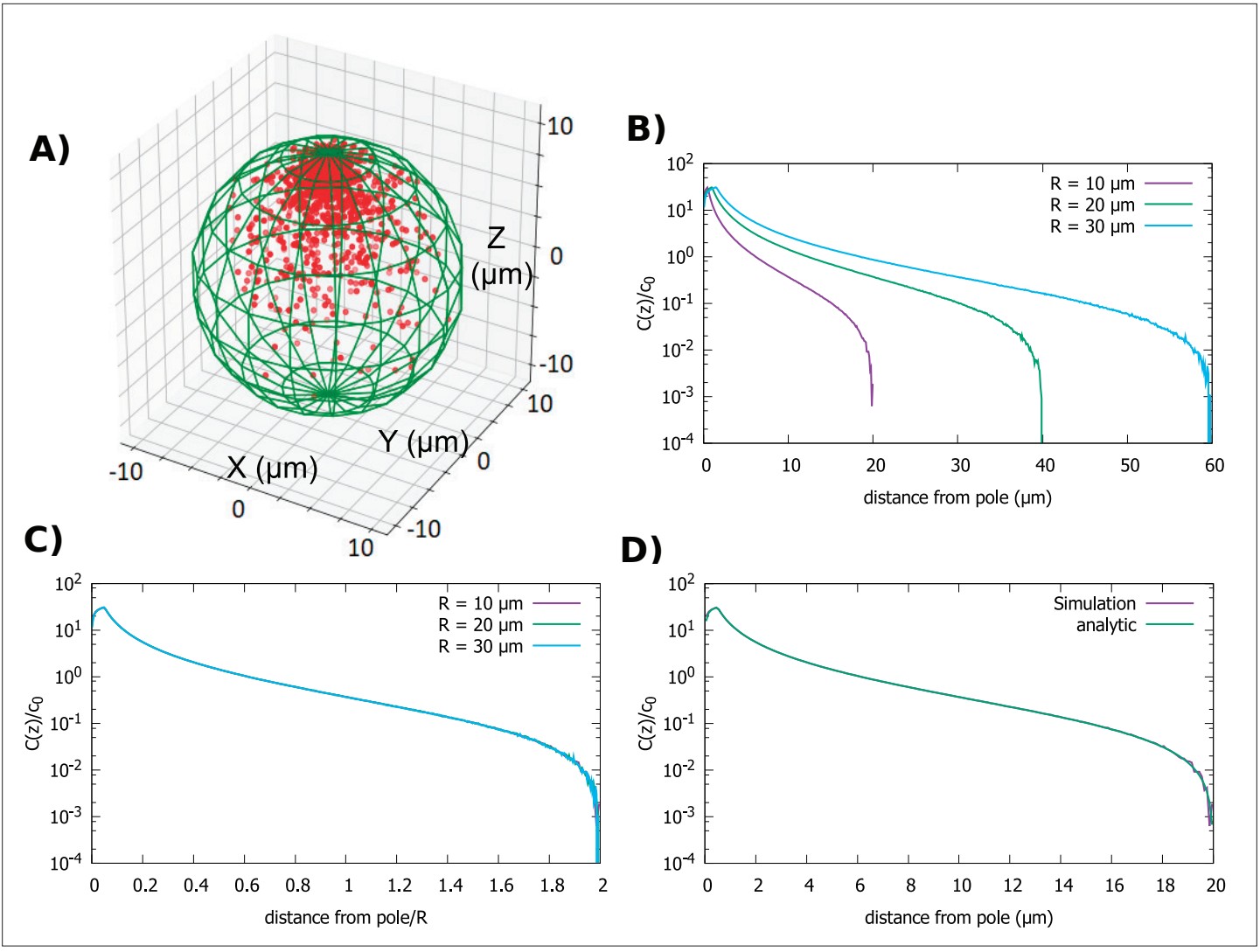

**Figure 2.** Concentration gradient in a spherical cell. (**A**) Sample steady-state configuration obtained from the direct simulation of the polar transport model in a spherical cell with 1000 proteins (red). Direction of polar transport on the surface of the cell is toward the anterior pole, where the accumulation of proteins is observed. (**B**) Concentration of proteins along the polar, $z$-axis normalized by the average cytoplasmic concentration, for spherical cells of different radii. (**C**) Concentration profiles from B scaled by the cell radius. (**D**) Comparison between the steady-state solution of the polar transport equations (*Equation 3*), compared to the protein concentration obtained from simulations, for a spherical cell with radius $10\ \mu m$. For all plots, the diffusion constant $D = 1\ \mu m^2/s$ and the transport speed along the cortex is $v = 1 \mu m/s$.

From this equation we can compute the concentration gradient along the polar, $z$-axis by averaging over the radial and angular coordinates:

$$C\left(z\right) = \frac{\int_0^{\sqrt{R^2-z^2}}\int_0^{2\pi} d\rho\, d\phi\, \rho c}{\pi\left(R^2-z^2\right)}$$

$$= \frac{2q}{R}\frac{\left[\left(\sqrt{1^2-\frac{z^2}{R^2}+\left(\frac{z}{R}-\frac{b}{R}\right)^2}-\left|\frac{z}{R}-\frac{b}{R}\right|\right)-\frac{a}{R}\left(\sqrt{1^2-\frac{z^2}{R^2}+\left(\frac{z}{R}-\frac{a}{R}\right)^2}-\left|\frac{z}{R}-\frac{a}{R}\right|\right)\right]}{4\pi\left(1-\frac{z^2}{R^2}\right)}, \tag{5}$$

where $b = R - \epsilon$, $a = \frac{R^2}{b}$ . If we define the small release distance from the pole $\epsilon$ such that is scales with $R$, then $\frac{b}{R}$ and $\frac{a}{R}$ are both constants, independent of $R$. Note that the choice of the release point only affects the gradient on distances away from the pole that are of order $\epsilon$ (i.e., the bump in *Figure 2D* at $z \sim \epsilon$) so a different choice of the release point will not disrupt the scaling property of the gradient in the bulk of the cell. Namely, at distances $z \gg \epsilon$, *Equation 5* can be expended in the small quantity $\epsilon/R$ and $\epsilon$ cancels out when we normalize the gradient by the average protein concentration.

Since, typically the average protein concentration ($c_0$) in cells across cell sizes is observed to be constant (*Milo and Phillips, 2015*), the quantity we are interested in is $C(z)/c_0$, where

$$c_0 = \frac{\int_{volume} c}{volume\ of\ the\ cell} = \frac{q}{8\pi R}\left[1 - \frac{b^2}{R^2}\right]. \tag{6}$$

In *Figure 2D*, we compare $C(z)/c_0$ obtained from combining *Equations 5 and 6* to the concentration gradient obtained from direct simulations of particle dynamics (see Materials and methods), and we see excellent agreement. Furthermore, by inspection of the analytic formulas, we see that $C(z)/c_0$ is not a function of $z$ alone but of the ratio $z/R$. This implies that the cytoplasmic gradient scales with cell radius. Therefore, the positional information conveyed by the gradient is the relative distance from the anterior pole of the cell, rather than the absolute distance.

## Gradients in cylindrical cells of different size and shape

Next, we investigate the properties of the gradient in cells that are cylindrical in shape, such as *Schizosaccharomyces pombe* (fission yeast). This geometry introduces an additional length scale into the problem since the cylinder is defined by its length and radius. In particular, we investigate how changes in cell length affect the protein gradient. As in the case of the spherical cell, we assume that molecules upon reaching the cell surface are captured and transported along the surface to the pole of the cell, as shown in *Figure 1*. Similar results are obtained if we assume that transport occurs along cytoskeleton filaments

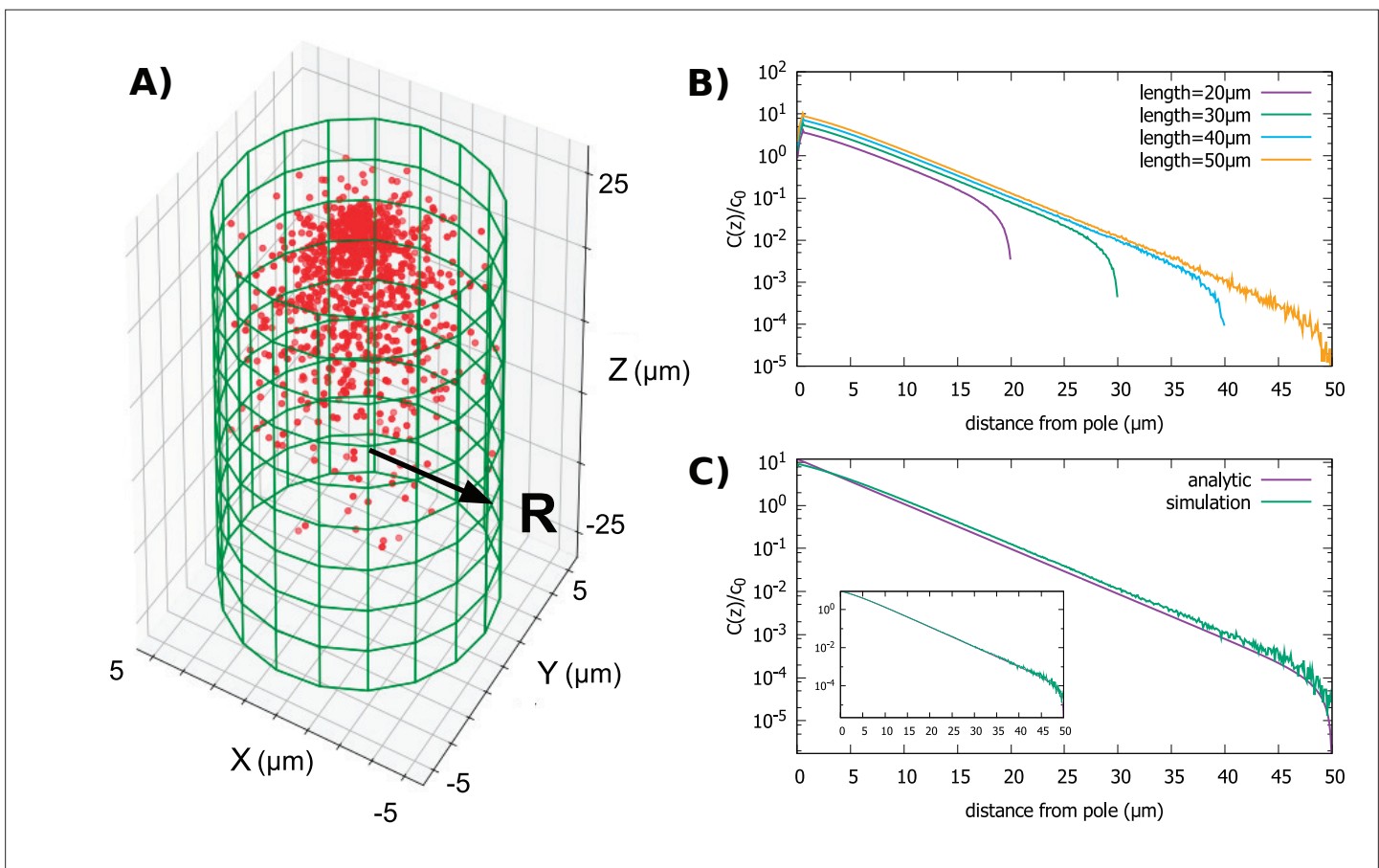

**Figure 3.** Protein gradient in a cylindrical cell. (**A**) Configuration of the proteins obtained from simulation of the polar transport model with 1000 proteins (red circles). Directed transport on the surface of the cell is along positive z-direction. (**B**) Protein concentration along the z axis, normalized by the average cytoplasmic concentration, for cylindrical cells of different lengths and fixed radius, $R = 10\mu m$. (**C**) Comparison between the steady-state solution of the polar transport equations and simulations. The analytic solution is an infinite series, and we compare the first term of this series to the results of simulations. The inset compares simulation results to the sum of the first 40 terms of the analytic solutions. For all plots, the diffusion constant $D = 1\ \mu m^2/s$ and the surface transport speed, $v = 1\mu m/s$.

extending along the central axis of the cylinder, as is the case of transport by myosin and kinesin motors in cilia. This case was examined in reference (*Naoz et al., 2008*) where the gradient of motor proteins was computed for a more detailed model of transport than included the rates of motors binding and unbinding from the filaments, but scaling of the concentration gradient with size was not investigated.

In *Figure 3A and B*, we show results from simulations of the polar transport model, where we vary the length of the cylinder while keeping the radius fixed. This is a biologically interesting case due to the fact that many cylindrically shaped cells, such as fission yeast (*Mitchison, 1985*), grow in this manner. The key observation we make from the simulations is that the gradient shape is roughly exponential, until it sharply falls off at the posterior pole of the cell; the decay length of the exponential is independent of the cell length, which only sets the extent of the gradient.

We confirm our observations from numerical simulations by solving analytically the steady-state concentration gradient (*Equation 3*) for the cylinder, by expanding the concentration into an infinite sum of Bessel functions (see Appendix A for details). This analytic solution is compared to simulation results in *Figure 3C*, inset. Furthermore, this infinite sum of Bessel functions is well approximated by the first term of the series

$$c\left(\rho,\ z\right) = \tfrac{1}{Z}\mathrm{J}_0\left(\tfrac{\rho}{\lambda}\right)\sinh\left(\tfrac{L+z}{\lambda}\right) \tag{7}$$

when $e^{2L/\lambda} \gg 1$; here, $\mathrm{J}_0$ is the Bessel function of order 0, $2L$ is the length of the cylinder, $\lambda$ is the decay constant of the gradient, $\rho = \sqrt{x^2 + y^2}$ is the polar coordinate, and $Z$ is a constant set by the total number of proteins in the cell. This approximation provides a simple formula for the decay constant $\lambda = R/2.4$, where 2.4 is the first root of $\mathrm{J}_0$. Clearly, the decay length $\lambda$ does not depend on the length of the cylinder and the shape of the gradient is mostly determined by the radius of the cell.

The fact that the decay length of the gradient is set by the radius of the cylinder leads to another interesting result. Namely, even if the average concentration of molecules in the cytoplasm $c_0 = \int c\left(\rho, z\right) / V_{cell}$ is kept fixed as the cell elongates and its volume ($V_{cell}$) increases, the concentration at the midpoint ($z = 0$) of the cell will decrease exponentially with the length of the cell as $\sim c_0 e^{-2.4L/R}$. Therefore, the polar transport mechanism provides the cell with the means to detect its overall length by locally monitoring the concentration of the protein species in the gradient, even when the total protein concentration is not changing with cell size. This protein gradient could be used by the cell to self-assemble a structure at a specified relative position away from the cell pole, like the midpoint, at a time in the cell cycle when the cell length reaches a particular threshold value.

## Gradients in spheroidal cells

The third shape that we examine is a spheroid, which roughly approximates the shape of a *Drosophila* embryo, or a budding yeast cell; see *Figure 4A*. Also, this shape is mathematically useful as it interpolates between the sphere and the cylinder. A spheroid is described by the equation: $\frac{x^2+y^2}{R^2} + \frac{z^2}{a^2} = 1$, where we assume $a \geq R$. When $a = R$ this equation describes a sphere, while for $a \gg R$ the shape resembles a cylinder. Indeed, our calculations of the gradient in this case show how the result obtained for the sphere morphs to that of the cylinder, as the long axis of the spheroid gets longer and longer.

First, we examine the situation when the cell size changes but the cell shape defined by the aspect ratio, $a/R$, is kept fixed. In this case a single length scale defines the size of the cell. In *Figure 4A and B* we show results of our particle simulations for this geometry. We observe that the gradient scales with the linear dimension of the cell. In *Figure 4C* we compare our analytic solution to results obtained from simulations. Like for the previous two cases the solution is obtained by solving *Equation 3*, which in this case can be written as an infinite sum of Legendre polynomials. (The full solution is given in Appendix B.)

*Figure 4D* shows how $C\left(z\right)/c_0$ changes as we make the cell more oblong while keeping the radius fixed. The shape of the gradient changes from that of a sphere ($a = R$) to one that resembles the cylinder case ($a \gg R$). In fact, the shape of the gradient in very oblong spheroids is well approximated by an exponentially decaying function of the distance from the pole (dashed line in *Figure 4D*), with a decay constant given by $R/2.4$, which is the result we obtained for a cylinder of radius $R$.

## Imperfect capture of proteins at the cortex

So far, we have only considered the case where a diffusing protein upon reaching the surface of the cell always gets taken up by a motor and is transported to the pole. In reality, we expect the binding

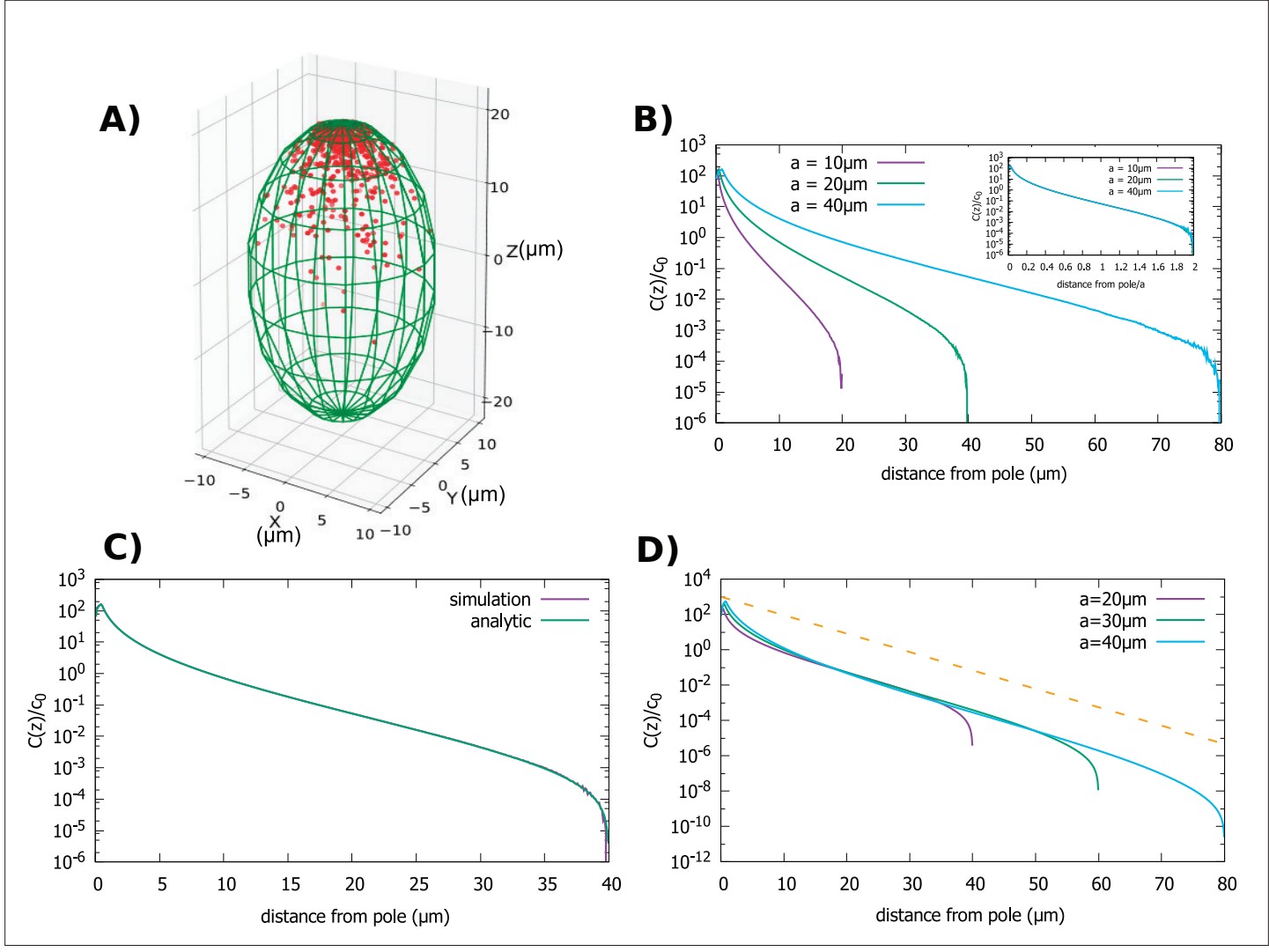

**Figure 4.** Concentration gradient in a spheroidal cell.
(**A**) Configuration of the proteins obtained from simulation of the polar transport model in a spheroidal cell of radius $R = 10\mu m$ and major axis $a = 20\mu m$, and 1000 proteins. Direction of transport on the surface of the cell is along the positive z direction. (**B**) Protein concentration normalized by the average cytoplasmic concentration for spheroidal cells of different radii, with a fixed aspect ratio $\frac{a}{R} = 2$. The inset shows concentration profiles where the distance from the pole is scaled by $a$. (**C**) Comparison between the analytic solution and simulations. (**D**) Concentration profiles for spheroids with the same minor axis ($R = 10\mu m$) and a varying major axis ($a$). The dashed line represents exponential decay with a decay constant $R/2.4$ as we computed above, for the case of a cylindrical cell. For all plots, the diffusion constant $D = 1\ \mu m^2/s$ and the transport speed along the cortex is $v = 1\mu m/s$.

of the protein to occur only occasionally, since no motor might be present at the point of impact of the protein on the surface. Also, even if the motor is present, not every collision between the protein and the motor will necessarily be productive leading to binding. To take this into account we reexamine our polar transport model by introducing a rate constant ($k_{on}$) that describes the binding of the proteins at the surface with motors moving along the filaments on the surface. This rate constant is the product of the second-order rate constant for proteins binding to motors and the number of motors per unit area of the cells surface (for details, see Appendix C). Therefore, $k_{on}$ has units of µm/s. If we incorporate this rate constant into our model, then at the cell boundary, the diffusive protein flux into the surface of the cell must match the protein uptake by the motors, which is described by a reactive boundary condition (**Erban and Chapman, 2007**),

$$-D\,\nabla c \cdot \hat{n}\big|_{at\ surface} = k_{on}c\big|_{at\ surface}\ ; \tag{8}$$

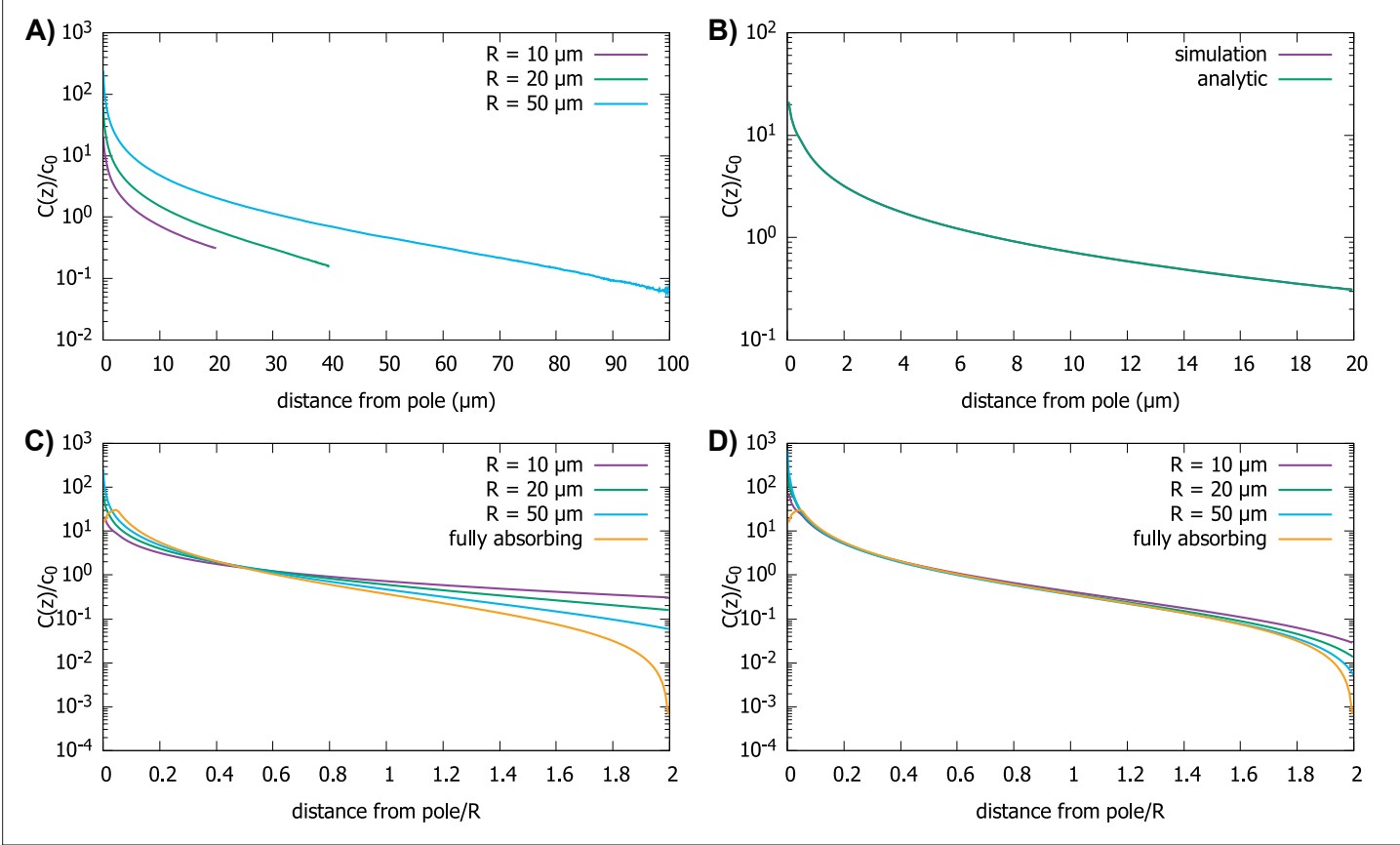

**Figure 5.** Effect of imperfect capture on the protein gradient.
(**A**) Protein concentration gradients (normalized by the average cytoplasmic concentration $c_0$) in spherical cells of different radii, computed by simulations of the polar transport model with 1000 particles. We use reactive boundary conditions with $k_{on} = 0.1\ \mu$ m/s. (**B**) Comparison of the simulated protein gradient in a spherical cell, $R = 10\ \mu$ m, with reactive boundary conditions ($k_{on} = 0.1\ \mu$ m/s), to the analytical solution for the concentration profile (see Appendix C). (**C**) Analytic results for concentration gradients for different cell radii plotted against the distance from the pole scaled by the radius; $k_{on} = 0.1\ \mu$ m/s. The scaling observed for fully absorbing boundary conditions (see *Figure 2C*) is absent. (**D**) Same as C, but now with $k_{on} = 1\ \mu$ m/s. Scaling is approximately restored for most distances away from the pole. For all plots, the diffusion constant $D = 1\ \mu$m²/s and the transport speed along the cortex is $v = 1\mu$m/s.

here, $\hat{n}$ is the normal to the cell surface. The fully absorbing boundary condition studied thus far corresponds to the limit $k_{on} \to \infty$, while for a fully reflecting surface, when there is no capture of proteins at the surface, $k_{on} = 0$. In the limit when the surface of the cell is reflective, there is of course no gradient and diffusion will lead to a constant concentration of proteins in the cell.

To analyze the effect of the reactive boundary condition on the protein gradient, we did simulations of the polar transport model in spherical cells of different radii; see *Figure 5A*. We also solved analytically *Equation 3* with this new boundary condition and compare it to the simulation results in *Figure 5B*. The solution is given as an infinite series of Legendre polynomials (see Appendix C), and we plot this solution for different parameter values in *Figure 5C and D*.

We find that the shape of the gradient is controlled by a dimensionless rate of capture, $k_{on}R/D$, where $R$ is the cell radius and $D$ is the protein diffusion constant. For small values of $k_{on}R/D$, which is the limit of reflecting boundary conditions, when the motors are not effective at capturing the proteins that arrive at the surface of the cell by diffusion, we find that the gradient flattens out and is no longer scale-invariant (see *Figure 5C*). On the other hand, for large values of the dimensionless rate of capture, $k_{on}R/D$, the shape of the gradient approaches that for a fully absorbing boundary condition and scaling is restored (see *Figure 5D*). In Appendix C we show that the condition $k_{on}R/D \gg 1$ corresponds to the condition that the number of motors on the surface, which can capture the diffusing proteins arriving at the surface, is larger than $R/b \approx 1000$, where $b$ is the size of the motor protein

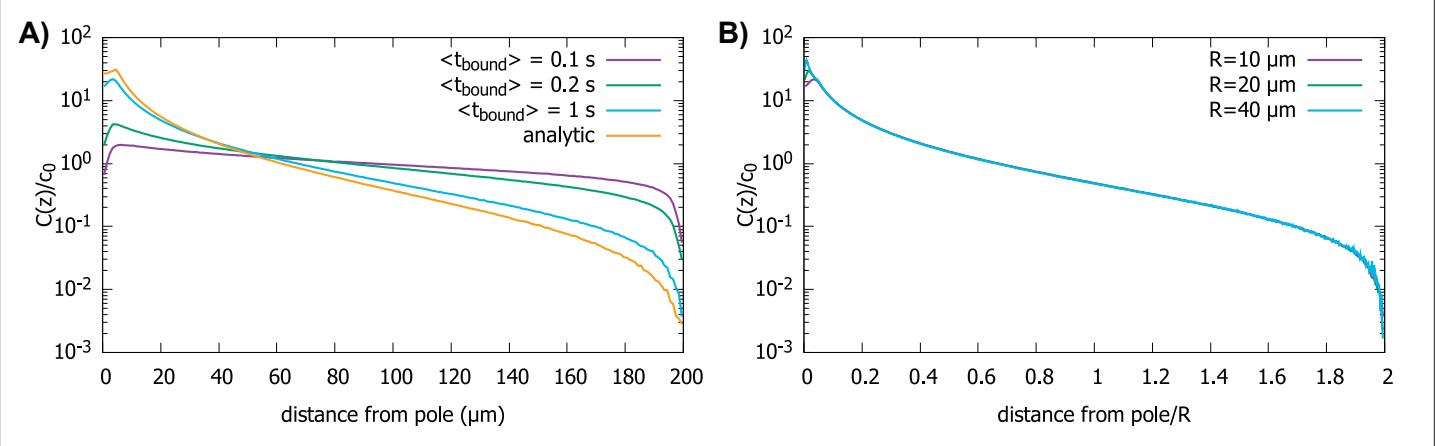

**Figure 6.** Effect of imperfect transport of proteins on the protein gradient. (A) Concentration gradient obtained from simulation with 1000 proteins for various values of $\langle t_{bound} \rangle$ with a spherical cell of radius $R = 10 \, \mu$m. All other parameters are the same as in *Figure 2*. The analytic curve represents the gradient obtained from the polar transport model for a spherical cell. (B) Concentration gradients in a scaled coordinate for different values of $R$ with $\langle t_{bound} \rangle = 1$ s. For all plots, the diffusion constant $D = 1 \, \mu$m$^2$/s and the transport speed along the cortex is.$v = 1 \mu$m/s.

(typically, a few nanometers). Note that this is a very small fraction of motors needed to cover the whole surface of the cell, which is approximately $4\pi R^2/b^2$ .

## Imperfect protein transport to the pole

In our model, we have assumed that once the protein molecules bind to the motors moving along the surface, they always get transported all the way to the anterior pole of the cell, where they are released. For example, this behavior is observed for proteins that take part in intraflagellar transport in *Chlamydomonas* (*Chien et al., 2017*). Still, there is always the possibility that the proteins will fall off the motor before the pole of the cell is reached. Therefore, next we consider the case of a perfectly reacting spherical cell where after a protein molecule is captured by a motor at the surface, it can detach into the cytoplasm with rate $\gamma$. For this unbinding to have an effect on the gradient, the average time the protein spends bound to the motor $\langle t_{bound} \rangle = 1/\gamma$ should be less than the time it takes for the transport to get it to the pole, $t_{transport} \sim R/v$, which is typically a few seconds.

In *Figure 6*, we show the simulation results for proteins diffusing in a spherical cell, captured at the surface by motors and transported to the pole, but now with a rate $\gamma$ for them falling off the motor. In *Figure 6A*, we show the effect of $\gamma$ on the protein gradient for the perfectly reactive spherical cell (*Figure 2D*). We see that, with decreasing $\langle t_{bound} \rangle = 1/\gamma$, the gradient flattens out, as would be expected for $\langle t_{bound} \rangle = 0$, since this maps to the perfectly reflective cell surface. In *Figure 6B*, we show that even though the presence of a finite residence time of the protein on the surface makes the gradient flatter, the scaling with the radius of the cell is preserved.

## Discussion

Herein, we analyze a model of gradient formation in cells that naturally leads to a protein gradient that scales with the linear dimensions of the cell. The key feature of the model is diffusion of proteins in the cytoplasm coupled to polar transport along the cell surface, which can be driven by motor transport, or cytoplasmic flow (*Howard et al., 2011*). In the case of a spherical cell, the gradient is approximately exponential with a decay length set by the cell's radius. In a cylindrically shaped cell, where the transport is along the long axis, the gradient is exponential with a decay length set by the radius, mostly independent of the length of the cell.

The emergence of a length scale, namely, the decay length of the protein gradient, which is due to the combined action of diffusion and advection, provides an alternative to the Turing mechanism, where a length scale is set by the diffusion constant and a reaction rate constant. Indeed, a number of recent papers have shown how the emergence of a length scale due to directed transport can lead to spatial patterns in concentration, which could be utilized by cells in the process of morphogenesis

(*Naoz et al., 2008*; *Howard et al., 2011*; *Hecht et al., 2009*; *Halatek et al., 2018*). Typically, the mechanisms described in these papers lead to a length scale that is set by parameters such as the speed of transport, the diffusion constant, and the rates of chemical reactions. What is unusual about the model discussed here is that the length scale of the gradient is independent of all these parameters and is set solely by the geometry of the cell.

Recently, such a gradient, one whose shape is defined by the shape of the compartment it forms in, was demonstrated in flagella of *Giardia* cells, where the directed transport of the non-motile kinesin-13 to the flagellar tips leads to an exponential gradient of kinesin-13. As predicted by our model, the decay length of the gradient was measured to be the same for all the flagella in *Giardia*, independent of their length, which varies between 7 and 14 µm. Furthermore, the decay length of the gradient is 0.4 µm, which is comparable to the radius of the flagella, as predicted by *Equation 7*.

While no experiments to date have directly probed the mechanism we propose for gradient formation in spherically shaped cells, there are many examples of such cells where directed transport is coupled to diffusion, in the manner assumed by our model. A promising candidate is the budding yeast cell, where the growth of the bud is enabled by a system of actin cables that is set up in the mother cell. The actin cables (typically, 10 of them) span the length of the cell and are localized to the cell cortex. This cable system can set up polar gradients via the directed transport of proteins by myosin V motors that move along the cables toward the bud while carrying secretory vesicles. In fact, a gradient of the protein Smy1, which is bound to secretory vesicles has been reported (*Eskin et al., 2016*), but their dependence on cell size has yet to be measured.

Another way that a polar gradient can form in budding yeast cell is by proteins bound to the side of the actin cables being transported away from the bud by the treadmilling action of actin filaments. Treadmilling is expected to occur here by the dual action of actin polymerization, which occurs at the bud neck by the activity of formin proteins localized to the septin ring, and depolymerization, which occurs toward the opposite pole of the cell (*Chesarone-Cataldo et al., 2011*).

To get a sense of the size of the protein gradients one can expect in yeast cells due to the treadmilling of actin cables, we can make a simple order of magnitude estimate. The measured yeast cable extension rate is about $0.5\frac{\mu m}{s}$ and there are about 10 cables per cell, each about 5 actin filaments thick (*Chesarone-Cataldo et al., 2011*). Given that about 300 actin monomers are in a micron of actin filament, this leads to an actin turnover rate of about $8 \times 10^3$ monomers/s, due to the assembly and disassembly of the actin cables. In steady state, this flux of actin from the bud neck (the anterior pole), to the posterior pole of the cell is balanced by the diffusive flux going in the opposite direction, which is generated by the higher concentration ($c_P$) of actin at the posterior pole than at the anterior ($c_A$). Estimating this diffusive flux by the expression, $D\frac{c_P-c_A}{2R}\pi R^2$ , where $D \approx 1\mu m^2/s$ is the diffusion constant for actin monomers, $R \approx 2\mu m$ is the radius of the yeast cell ($\pi R^2$ being the cross-sectional area), yields a concentration difference between the two poles $c_P - c_A \approx 3\mu M$.

Another intriguing possibility for experimentally measuring the polar gradient described by our model is provided by mRNA localization, which has been observed in many different cell types (for a review, see *Holt and Bullock, 2009*). Subcellular localization of mRNA is typically achieved by polar active transport of mRNA and localized anchoring at the cell pole. In some instances, like in the case of the *bicoid* mRNA localization in late oocytes of the fruit fly, the anchors are not persistent over relevant time scales and localization is achieved by continual active transport to the pole, in this case by dynein motors moving along microtubules (*Weil et al., 2006*). Here the localization of mRNA satisfies the key assumptions of our model, polar transport and dynamic attachment at the pole, that, we predict, can lead to a scale-invariant mRNA gradient. Note that, if the protein that is produced by translating the mRNA does not diffuse substantially over its lifetime, the scaling of the mRNA gradient will be imprinted on the related protein gradient.

When defining the polar transport model (*Figure 1*), we described the transport of proteins as being due to molecular motors moving along filaments that have a polar arrangement. Instead of directed motion due to motors, advection of proteins to the pole of the cell could also be achieved by cytoplasmic flows (*Howard et al., 2011*). Furthermore, for our results to hold, directed transport to the pole is not required. The key ingredient, which is encoded in *Equation 3*, is that the cell pole serves as a source and the surface of the cell as the sink for proteins diffusing in the cytoplasm, while the total number of proteins stays fixed. For example, diffusion of proteins along the surface of the cell will do just as well as advection, as long as the proteins, once bound and diffusing on the surface,

can only detach at one of the poles (e.g., due to an interaction with another factor localized at the cell pole). Another possibility is that proteins only experience diffusive transport in the cytoplasm but get 'activated' at the cell pole and 'deactivated' by an interaction with the cell surface. Again, in this case, the concentration of active protein in the cytoplasm will be given by *Equation 3*, and will therefore form a scale-invariant gradient of activity in steady state.

As described in the Introduction, a scale-invariant protein gradient is useful to a cell that is in need of determining position within the cell relative to its linear dimensions, such as in the case of patterning in development (*Gregor et al., 2005*). Another possible use for such a gradient could be in controlling the self-assembly of linear structures such as actin cables and flagella, which are precisely scaled to the linear dimensions of the cell (*McInally et al., 2019*; *Bauer et al., 2021*; *Datta et al., 2020*). For example, a recent study of actin cables in yeast found that when cells were mutated to grow abnormally large, the actin cables assembled to a length that was proportional to the cell radius (*McInally et al., 2021*). A hypothesized explanation of this phenomena is the presence of a scale-invariant gradient of depolymerizing activity. An intriguing possibility for the observed scaling of actin cable lengths with cell length is that cable dependent polar transport of depolymerizing proteins leads to their gradient and subsequent control of the length of the cables themselves.

Given the ubiquity of cortical transport in a variety of cells, we suspect that the mechanism of gradient formation described here could be in wide use. We hope that our theoretical results will provide impetus for experiments that will identify such gradients and test the proposal that self-assembly of structures that scale with the linear dimensions of the cell use such gradients as a read-out of the cell length.

## Materials and methods

### Simulation details

All the simulations start at time $t = 0$ with $N$ protein molecules uniformly distributed inside the cell, whose shape is either a sphere, a cylinder, or a spheroid. For the diffusion constant of the proteins in the cytoplasm, we take $D = 1 \mu\text{m}^2/\text{s}$, while for the speed of transport we take $v = 1\ \mu m/s$, values typical for cytoplasmic diffusion and motor transport in cells. In each simulation step, we increase the time by $\Delta t$, usually taken to be $0.001 s$, and the position of the $i$th protein, if it is in the cytoplasm, is updated according to the following rule:

$$r_i (t + \Delta t) = r_i (t) + \xi_i. \tag{9}$$

Here, $\xi_i$ is a Gaussian random variable with $\langle \xi_{i\alpha} \rangle = 0$ and $\langle \xi_{i\alpha} \xi_{i\alpha'} \rangle = \sqrt{2D\Delta t}\delta_{\alpha,\alpha'}$ ; $\alpha$ denotes the $x, y, z$-components of the position vector.

When a protein molecule hits the surface of the cell, it gets taken up by the polar transport. If the point on the cell surface at which it is taken up by the transport is $r_c = (x_c, y_c, z_c)$ , then the distance the protein has to travel to get to anterior pole is $d(z_c)$. The time to reach anterior pole, or equivalently, the time the protein molecule spends on the surface is, $t_s = d(z_c)/v$. So, after a molecule is taken up by the polar transport, it is released from the release point $r_0$ into the cytoplasm of the cell after time $t_s$ , or after $(t_s/\Delta t)$ simulation steps.

The distance to the anterior pole can be computed analytically for each of the cell shapes that we consider:

- For a spherical cell, $d(z_c) = R\cos^{-1}(z_c/R)$.
- For cylindrical cell, $d(z_c) = L - z_c$ .
- For spheroidal cell, $d(z_c) = a(E(\pi/2, k) - E(\varphi, k))$ , where $\varphi = \sin^{-1}(z_c/a)$ , $k = \sqrt{1 - R^2/a^2}$, and $E(\varphi, k)$ is elliptic integral of the second kind.

We run the simulation until the protein gradient has reached a steady state. To compute $C(z)$ the steady-state cytoplasmic protein gradient along the polar, $z$-axis, we split the cell into thin slices along the $z$-axis, compute the number of proteins in a given slice at position $z$, and divide by the volume of the slice. The average cytoplasmic protein concentration $c_0$ is the total number of proteins in the cytoplasm divided by the volume.

## Acknowledgements

This work was supported by the National Science Foundation grants DMR-1610737 and MRSEC DMR-2011846, and by the Simons Foundation. We thank Ariel Amir, Rob Phillips, Thomas Fai, Shane McInally, and Apoorv Umang for useful discussions and comments on the manuscript. SG thanks NCBS, Bengaluru for their hospitality.

## Additional information

### Funding

| Funder | Grant reference number | Author |
|---|---|---|
| National Science Foundation | DMR-1610737 | Arnab Datta Sagnik Ghosh Jane Kondev |
| Brandeis University | MRSEC - DMR-548 2011846 | Arnab Datta Sagnik Ghosh Jane Kondev |
| Simons Foundation | | Arnab Datta Sagnik Ghosh Jane Kondev |

The funders had no role in study design, data collection and interpretation, or the decision to submit the work for publication.

### Author contributions

Arnab Datta, Sagnik Ghosh, Conceptualization, Data curation, Formal analysis, Investigation, Methodology, Project administration, Resources, Software, Validation, Visualization, Writing – original draft, Writing – review and editing; Jane Kondev, Conceptualization, Funding acquisition, Methodology, Project administration, Resources, Supervision, Writing – original draft, Writing – review and editing

### Author ORCIDs

Arnab Datta http://orcid.org/0000-0002-7474-1720
Sagnik Ghosh http://orcid.org/0000-0001-7174-8479
Jane Kondev http://orcid.org/0000-0001-7522-7144

### Decision letter and Author response

Decision letter https://doi.org/10.7554/eLife.71365.sa1
Author response https://doi.org/10.7554/eLife.71365.sa2

## Additional files

### Supplementary files
• Transparent reporting form

### Data availability

All data generated or analysed during this study are included in the manuscript and supporting files. Source Code is available at https://github.com/gnick08/cell-gradients (copy archived at swh:1:rev:71e48be3a2e264fdb872fe95acd6e5d84b919bf0).

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

# Appendix 1

## A. Cylindrical cell

For the cylindrical cell, *Equation 3* can be solved using separation of variables (*Mohammad et al., 2015*) and is given by:

$$c\left(\boldsymbol{r}\right) = \sum_{n=1}^{\infty} \frac{q}{\pi R x_{0n}\left[J_1\left(x_{0n}\right)\right]^2} J_0\left(\frac{x_{0n}\rho}{R}\right) \frac{\sinh\left(\frac{x_{0n}(L-z_>)}{R}\right)\sinh\left(\frac{x_{0n}(L+z_<)}{R}\right)}{\sinh\left(\frac{2x_{0n}L}{R}\right)},$$ (A1)

where, $z_> = \max\left(b, z\right)$, $z_< = \min\left(b, z\right)$, $b = L - \epsilon$, $J_0$ and $J_1$ are Bessel function of order 0 and 1, respectively, and $x_{0n}$ is the $n$ th root of $J_0$.

*Equation 4* is the first term in *Equation A1* and it provides a good approximation of the full series (see *Figure 3C*).

To calculate $C\left(z\right)$ from $c\left(\boldsymbol{r}\right)$ in *Equation A1*, we have to calculate the integral, $\int_0^R J_0\left(\frac{x_{0n}\rho}{R}\right) \rho \, d\rho$. This we do using the recurrence relation for the Bessel function, $x J_0\left(x\right) = \frac{d\left(x J_1\left(x\right)\right)}{dx}$, which gives:

$$C\left(z\right) = \frac{\int_0^R \int_0^{2\pi} d\rho \, d\phi \, \rho c}{\pi R^2} = \sum_{n=1}^{\infty} \frac{2q}{\pi R x_{0n}^2 J_1\left(x_{0n}\right)} \frac{\sinh\left(\frac{x_{0n}(L-z_>)}{R}\right)\sinh\left(\frac{x_{0n}(L+z_<)}{R}\right)}{\sinh\left(\frac{2x_{0n}L}{R}\right)}.$$ (A2)

## B. Spheroidal cell

For spheroidal cell, we again solve *Equation 3* with $\boldsymbol{r}_0 = \left(0, 0, a - \epsilon\right) = \left(0, 0, b\right)$ using the prolate spheroid co-ordinate $\left(\xi, \eta, \phi\right)$:

$$x = f\sqrt{\left(\xi^2 - 1\right)\left(1 - \eta^2\right)}\cos\phi$$

$$y = f\sqrt{\left(\xi^2 - 1\right)\left(1 - \eta^2\right)}\sin\phi$$

$$z = f\xi\eta,$$

where $f = \sqrt{a^2 - R^2}$, $1 \leq \xi < \infty$, $-1 \leq \eta \leq 1$, $0 \leq \phi < 2\pi$. For all points on the surface of our spheroidal cell $\xi$ stays constant and we denote it by $\xi_s = a/f$. The coordinates of the release point in the vicinity of the cell pole are given by: $\left(\xi_b = \max\left(b/f, 1\right), \ \eta_b = \min\left(b/f, 1\right), 0\right)$.

The solution for the steady-state cytoplasmic concentration is given as a sum of the inhomogeneous solution and homogeneous solutions of *Equation 3*, with the coefficients chosen to satisfy the boundary condition (*Xue and Deng, 2017*),

$$c\left(\boldsymbol{r}\right) = \frac{q}{4\pi\left|\boldsymbol{r} - \boldsymbol{r}_0\right|} + \sum_{n=0}^{\infty} A_n P_n\left(\xi\right) P_n\left(\eta\right),$$ (B1)

where

$$A_n = -\frac{q}{4\pi f} \frac{(2n+1)P_n\left(\xi_b\right)Q_n\left(\xi_s\right)}{P_n\left(\xi_s\right)},$$

where $P_n$ is the Legendre polynomial of order $n$ and $Q_n$ is the Legendre function of second kind of order $n$.

The concentration profile along the polar axis of the cell is given by the integral

$$C\left(z\right) = \frac{2\int_0^{R\sqrt{1 - z^2/a^2}} d\rho \, \rho \, c\left(\boldsymbol{r}\right)}{R^2\left(1 - z^2/a^2\right)},$$ (B2)

which we integrate numerically to obtain the results shown in *Figure 4C, D*.

## C. Semi-absorbing spherical cell

We use the same notation as in the spherical cell section. The solution of *Equation 3* can be written as:

$$c\left(\mathbf{r}\right) = \frac{q}{4\pi\left|\mathbf{r}-\mathbf{r_0}\right|} + \sum_{n=0}^{\infty} A_n r^n \mathrm{P}_n\left(\cos\theta\right),$$ (C1)

with $A_n$ to be determined. At the surface $r = R > b$. So, we use the expansion:

$$\frac{q}{4\pi\left|\mathbf{r}-\mathbf{r_0}\right|} = \sum_{n=0}^{\infty} \frac{q}{4\pi} \frac{b^n}{r^{n+1}} \mathrm{P}_n\left(\cos\theta\right).$$

For the spherical cell, the boundary condition (*Equation 8*) becomes:

$$-D\frac{\partial c}{\partial r} = k_{on}c.$$ (C2)

Then, matching the coefficient of $\mathrm{P}_n\left(\cos\theta\right)$ at the surface, we get:

$$A_n = \frac{q}{4\pi} \frac{b^n}{R^{2n+1}} \frac{n+1-\frac{k_{on}R}{D}}{n+\frac{k_{on}R}{D}}.$$ (C3)

To estimate the dimensionless number $k_{on}R/D$, we assume that on the surface of the cell, the motors are circular absorbers of radius $r_a$ and the surface concentration of the absorbers is $c_{surface}$.

If we assume the reaction of a single motor and the cytoplasmic protein is diffusion limited, then the reaction rate for motors taking up proteins is given by $4\pi fDr_a$. Here, $f < 1$ accounts for the fraction of diffusion-driven encounters that lead to productive binding of the protein to the motor. At any point on the surface, the number of binding reactions per unit time is given by:

$$k_{on}c = 4\pi fDr_a c_{surface}c.$$ (C4)

where $c$ is the concentration of cytoplasmic protein at the surface of the cell.
Therefore,

$$\frac{k_{on}R}{D} = 4\pi fr_aRc_{surface}.$$ (C5)

If the surface of the cell was fully covered with motor proteins, we would have $c_{surface} \approx 1/r_a^2$. To account for the coverage not being 100%, we write $c_{surface} = f_{surface}/r_a^2$, where $f_{surface} < 1$. Therefore,

$$\frac{k_{on}R}{D} = 4\pi ff_{surface}\frac{R}{r_a}.$$ (C6)

Since the size of the motors is of a few nanometers, $R/r_a \sim 1000$. Putting this in *Equation C6*, we arrive at the estimate

$$\frac{k_{on}R}{D} \sim 10^4 ff_{surface}.$$ (C7)

If we assume the product $ff_{surface}$ is not smaller than say $10^{-2}$ then $\frac{k_{on}R}{D} \gg 1$, and the gradient produced will still be to a good approximation scale invariant (see *Figure 5D*).

## D. Surface concentration

In addition to the cytoplasmic protein concentration, which has been the focus of our study, one can also ask about the steady-state concentration on the surface of the cell. From *Equation 2.3*, which relates the surface concentration to the cytoplasmic concentration, we can infer that the scaling property of the cytoplasmic concentration will also hold for the surface concentration. To confirm this analytic result, we plot results of our simulations of the polar transport model for a spherical cell in *Appendix 1—figure 1*, which shows that the surface concentrations for cells of different radii collapse onto a single curve, when the distance from the anterior pole is scaled by the radius of the cell. To demonstrate this scaling property, we plot, $C_s\left(z\right)$, the average surface concentration along a thin ring along the surface, at distance $z$ away from the anterior pole, scaled by $c_{s,0}$, the average surface concentration, as a function of $z/R$.

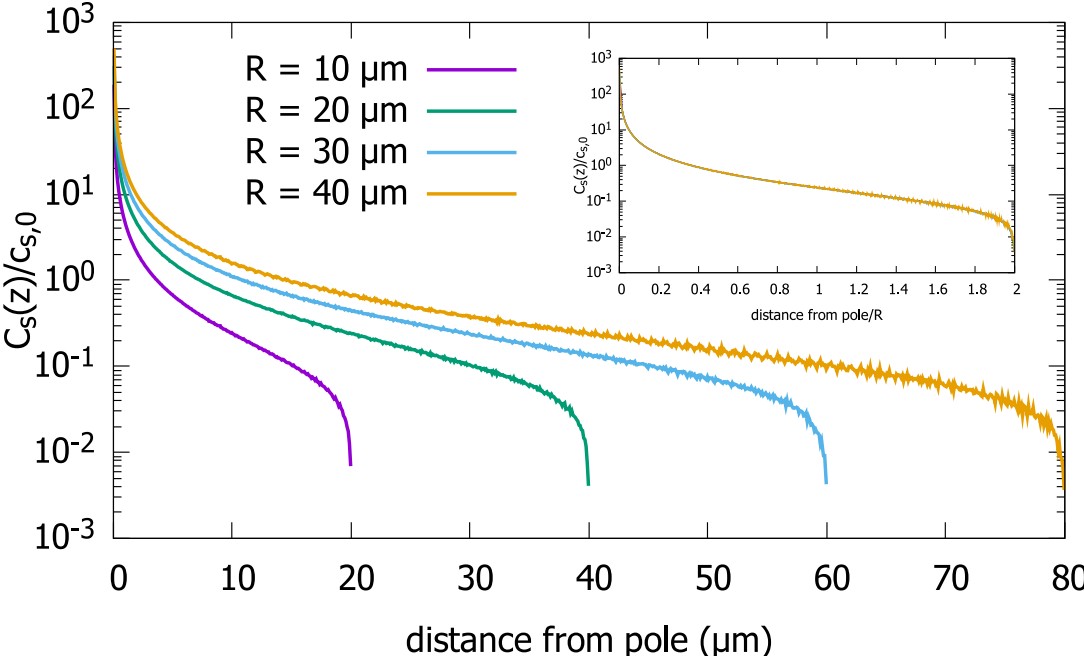

**Appendix 1—figure 1.** Surface concentration gradient for a spherical cell. Concentration profiles on the surface of spherical cells of different radii, obtained from simulations of the polar transport model of gradient formations. Inset shows the same data, just with the distance from the pole scaled by the cell radius. Parameters used in the simulation: protein diffusion constant $D = 1\mu\text{m}^2/\text{s}$, transport speed along the cortex $v = 1\mu\text{m/s}$, and $\epsilon/R = 0.05$ is the release point.

