## [Editor Report]

How biological patterns such as concentration gradient scale with the size of the cell or organism is a long-standing question in developmental and cell biology. In this study, Datta et al., show theoretically that directed membrane transport of biomolecules and their release at the cell pole results in a cytoplasmic gradient that scales with cell size if two requirements are met: the cell grows while maintaining its spheroid proportions, (i.e. not by elongation), and the binding of the cytoplasmic fraction of the biomolecule to the membrane should be close to irreversible. While there are no experiments available to date to directly probe the proposed mechanism, it could be achieved through several biochemical implementations and can inspire experimental studies.

---

## [Decision Letter]

**Decision letter after peer review:**

Thank you for submitting your article "How to Assemble a Scale-Invariant Gradient" for consideration by *eLife*. Your article has been reviewed by 2 peer reviewers, and the evaluation has been overseen by a Reviewing Editor and Naama Barkai as the Senior Editor. The reviewers have opted to remain anonymous.

Essential revisions:

As you will see from the reports below, the reviewers found your theoretical study potentially interesting. However, one main concern is the lack of comparison with experiments. Given the relative simplicity of the proposed mechanism, much more thorough experimental test should be feasible, and would considerably strengthen the paper. In addition, the reviewers provide a number of technical queries that need to be answered. These include:

1) Thorough comparison with experiments.

2) Role of the boundary conditions. As we are looking for a system's size scaling, it does not seem fair to also scale the short length cutoff (emission length epsilon). How are the results modify if this length is kept constant (Reviewer #2, public review). Also discuss the importance of the absorbing boundary condition (Reviewer #2, Recommendation and reviewer # 1 question 4).

3) Discussion the concentration distribution in the x,y plane (reviewer # 1 question 4).

4) Why are you only interested in concentration gradient in the cytoplasm? Many important processes depend on the membrane concentration. Provide a discussion of the membrane concentration as well.

*Reviewer #1 (Recommendations for the authors):*

Congratulations on this interesting study.

The following are a few suggestions that may improve this study. For the biological examples, it will be wonderful if the authors can analyze published data on e.g. bicoid or *C. elegans* and yeast ring. If this is not possible, the authors should be careful in suggesting their model may explain phenomena in specific biological systems.

Some suggestions on the theory:

1. A sensitivity analysis would be useful to understand how sensitive is the gradient and its scaling property to parameters such as c0, Kon, R (is there a limit?), R_0 (source size) and geometries (changes in L/R ratio).

2. A parameter screen that involves changing several parameters at a time (e.g. on spherical geometry) and testing scaling would inform of some interactions between parameters and provide the ranges where this mechanism may work or not, and not the boundaries which are more easily understood analytically.

3. For the analysis, it would be interesting to compare the k_on_ with the speed of transport in the membrane. Is there an interaction there? Would fast transport allow scaling for lower Kon?

4. In the cylinder, does the gradient scale along the radius of the cell (x-y plane)? It can be an interesting result. The gradient may be linear? Again for the cylinder, a boundary condition at the opposite pole=0, together with rapid diffusion or no degradation (e.g. reflective boundaries along the membrane) which result in a long length scale will yield a linear gradient that scales in the z-axis. This is perhaps trivial, but maybe there is a geometry/parameter interaction where a low k_on_ can result in scaling in this geometry specifically. This is only my intuition.

The point in the text where you argue the mechanism can be useful for cells is in my opinion weak and does not support the overall claim.

5. What is the timescale to form a gradient? It should be faster than the protein degradation rate to support exclusion of turnover dynamics (with e.g. uniform production in the cell).

6. You mention R/b~1000, for a size of a motor. If its size is 10nm, the cell radius is 10um, which is quite big for a unicellular organism (more like a mammalian cell), so please check that the number make sense in your argument?

*Reviewer #2 (Recommendations for the authors):*

Please specify more clearly the boundary conditions at the flat surface of the cylinder in Figure 3. Do the results of this figure depend on not having an absorbing surface near the anterior pole but only along the cylindrical part?

---

## [Author Response]

Essential revisions:As you will see from the reports below, the reviewers found your theoretical study potentially interesting. However, one main concern is the lack of comparison with experiments. Given the relative simplicity of the proposed mechanism, much more thorough experimental test should be feasible, and would considerably strengthen the paper. In addition, the reviewers provide a number of technical queries that need to be answered. These include:1) Thorough comparison with experiments.

While there are no experiments that we are aware of that directly probe the mechanism we propose for gradient formation, there are many examples of cells where directed transport of proteins is coupled to diffusion, which are the two key ingredients of the polar transport model we investigate theoretically. At the single cell level, a promising candidate for observing the protein gradient we have described are budding yeast cells. In budding yeast cells, the growth of the bud is enabled by a system of actin cables that is set up in the mother cell. The actin cables (typically, 10 of them, each about 5 μm long) span the length of the cell and are localized to the cell cortex. This cable system can set up polar gradients via the directed transport of proteins by myosin V motors that move along the cables toward the bud while carrying secretory vesicles. In fact, a gradient of the protein Smy1, which is bound to secretory vesicles was reported by the Goode lab [Eskin J, et al., 2016, Mol Biol Cell 27(5):828-37], but their dependence on cell size has yet to be measured.

Another way that a polar gradient can form in budding yeast cell is by having proteins bound to the side of the actin filaments and transported away from the bud by the treadmilling action of actin filaments. Treadmilling is expected to occur here by the dual action of actin polymerization, which occurs at the bud neck and is carried out by formin proteins localized to the septin ring, and depolymerization, which occurs toward the opposite pole of the cell. The existence of a polar gradient in budding yeast that scales with cell size, if confirmed, could be the key to understanding the observed scaling of the septin ring and of the actin cables with cell size, as we’ve described in the introduction.

To get a sense of the size of the protein gradients one expects in yeast cells due to the treadmilling of actin cables, we can make a simple order of magnitude estimate. The measured yeast cable extension rate is about 0.5μmsec there are about 10 cables per cell, each about 5 actin filaments thick [Chesarone-Cataldo et al., (2011), Dev Cell 21: 217-230]. Given that about 300 actin monomers are in a micron of actin filament, this all gives an actin turnover rate of about 8×103 monomers/sec, for all the cortical cables. In steady state, this flux of actin from the bud neck (the anterior pole), to the posterior pole of the cell is balanced by the diffusive flux going in the opposite direction, which is generated by the higher concentration (cP) of actin at the posterior pole than at the anterior (cA). Estimating this diffusive flux by the expression, DcP−cA2RπR2, where D≈1μm2/sec is the diffusion constant for actin monomers, R≈2μm is the radius of the yeast cell, yields a concentration difference between the two poles cP−cA≈3 μM, which is typical of yeast proteins in the cytosol.

Another intriguing possibility for experimentally measuring the gradient described in our manuscript, is provided by mRNA localization that has been observed in many different cell types [for a review see, C.E. Holt and S.L. Bullock, Science 326, 1212-1216, 2009]. Subcellular localization of mRNA is typically achieved by polar active transport of mRNA and localized anchoring, via proteins it is in complex with, at the cell pole. In some instances, like in the case of the bicoid mRNA localization on late oocytes, the anchors are not persistent over relevant time scales and localization is achieved by continual active transport to the pole, in this case by dynein motor proteins [T.T. Weil et al., Dev. Cell 11, 251 (2006)]. In this case, the localization of mRNA satisfies the key assumptions of our model, polar transport and dynamic attachment at the pole, that we predict can lead to a scale invariant mRNA gradient. Note that, if the protein that is made by translating the mRNA does not diffuse substantially (more than over distances comparable to the cell size) over its lifetime, the scaling of the mRNA gradient will be imprinted on the protein gradient as well.

We have added this discussion of budding yeast and mRNA localization in embryos as possible realizations of the polar transport model of gradient formation into the Discussion section of the manuscript.

2) Role of the boundary conditions. As we are looking for a system's size scaling, it does not seem fair to also scale the short length cutoff (emission length epsilon). How are the results modify if this length is kept constant (Reviewer #2, public review). Also discuss the importance of the absorbing boundary condition (Reviewer #2, Recommendation and reviewer # 1 question 4).

As suggested by the referee, we did a careful analysis of the effect of the choice of the distance of the release point (i.e., the source) from the pole on the gradient. In Author response image 1 we see that as we change the distance from the pole to the source, the concentration profile is only affected at distances from the pole that are of order ε, with a local maximum at that distance equal to ε. However, the concentration in the bulk remains unaffected. This independence from choice of ε also follows from equation 5, which is the analytic solution for the steady state gradient in a spherical cell. Namely, for ϵ=0 , c(r)=0, as proteins released at the north pole are immediately reabsorbed. If we expand the general solution for c(r) in the small quantity ϵ/R around ϵ=0, the resulting approximation is linear with ε and hence C(z)/c0 will be independent of ε , as confirmed by our simulations (Author response image 1).

In the manuscript we have added the analysis of equation 5 that shows that the choice of the distance from the pole to the source (ε does not affect the gradient except, possibly at distances of order ε).

**Author response image 1. sa2fig1:** Concentration gradient for cells with the source at different distances from the pole (ε). Concentration profiles with differing source points. We start very close to the pole and move further away. The radius of the sphere is 10 μm, the diffusion constant D=1 μm2/s and the transport speed along the cortex is v=1μm/s.

3) Discussion the concentration distribution in the x,y plane (reviewer # 1 question 4).

While we did not focus in our paper on the shape of the gradient in the direction perpendicular to the polar axis, this property of the steady state concentration can also be extracted from our analytic formulas (Equation 5 and 7). Author response image 2 shows how the concentration changes across the XY plane when z is kept fixed, for the case of the spherical cell. We see that the concentration initially decreases relatively slowly. As we approach the boundary, the concentration starts decreasing rapidly and goes to zero, due to the absorbing boundary condition at the boundary. Note that the decay profile depends on the geometry of the cell. Unlike the example of the spherical cell plotted in Author response image 2, in a cylindrical cell the concentration along the direction perpendicular to the polar axis is a Bessel function of order zero (equation 7) which decays more rapidly than what is observed in a sphere (Author response image 2).

**Author response image 2. sa2fig2:** Concentration gradient in XY plane for a spherical cell. Concentration Profiles in the XY plane for three different distances from the pole. The radius of the sphere is 10 μm, the diffusion constant D=1 μm2/s and the transport speed along the cortex is v=1μm/s.

4) Why are you only interested in concentration gradient in the cytoplasm? Many important processes depend on the membrane concentration. Provide a discussion of the membrane concentration as well.

Following the suggestion of the referee, we analyzed the scaling property of the surface concentration of the protein in a spherical cell. We found, that like the bulk concentration, the concentration on the surface scales with the radius of the cells (Author response image 3). As the reviewer suggests, this could also be exploited by cells as it provides positional information along the cell surface. We have added these results as a new section in the Appendix.

**Author response image 3. sa2fig3:** Surface concentration gradient for a spherical cell.